# Combining genomics and epidemiology to track mumps virus transmission in the United States

Shirlee Wohl[1,2☯*], Hayden C. Metsky[1,3☯], Stephen F. Schaffner[1,2,4☯], Anne Piantadosi[1,5], Meagan Burns[6], Joseph A. Lewnard[7], Bridget Chak[1,2], Lydia A. Krasilnikova[1,2], Katherine J. Siddle[1,2], Christian B. Matranga[1], Bettina Bankamp[8], Scott Hennigan[6], Brandon Sabina[6], Elizabeth H. Byrne[1,2], Rebecca J. McNall[8], Rickey R. Shah[1,2], James Qu[1], Daniel J. Park[1], Soheyla Gharib[9], Susan Fitzgerald[9], Paul Barreira[9], Stephen Fleming[6], Susan Lett[6], Paul A. Rota[8‡], Lawrence C. Madoff[6,10‡], Nathan L. Yozwiak[1,2‡], Bronwyn L. MacInnis[1,4‡*], Sandra Smole[6‡], Yonatan H. Grad[4,11,12‡], Pardis C. Sabeti[1,2,4,13‡]

1 Broad Institute of MIT and Harvard, Cambridge, Massachusetts, United States of America, 2 Center for Systems Biology, Department of Organismic and Evolutionary Biology, Harvard University, Cambridge, Massachusetts, United States of America, 3 Department of Electrical Engineering and Computer Science, Massachusetts Institute of Technology, Cambridge, Massachusetts, United States of America, 4 Department of Immunology and Infectious Diseases, Harvard TH Chan School of Public Health, Boston, Massachusetts, United States of America, 5 Division of Infectious Diseases, Department of Medicine, Massachusetts General Hospital, Boston, Massachusetts, United States of America, 6 Massachusetts Department of Public Health, Jamaica Plain, Massachusetts, United States of America, 7 Division of Epidemiology and Biostatistics, School of Public Health, University of California, Berkeley, Berkeley, California, United States of America, 8 Division of Viral Diseases, Centers for Disease Control and Prevention, Atlanta, Georgia, United States of America, 9 Harvard University Health Services, Harvard University, Cambridge, Massachusetts, United States of America, 10 Department of Medicine, University of Massachusetts Medical School, Worcester, Massachusetts, United States of America, 11 Center for Communicable Disease Dynamics, Harvard TH Chan School of Public Health, Boston, Massachusetts, United States of America, 12 Division of Infectious Diseases, Brigham and Women's Hospital, Harvard Medical School, Boston, Massachusetts, United States of America, 13 Howard Hughes Medical Institute, Chevy Chase, Maryland, United States of America

☯ These authors contributed equally to this work.
‡ These authors jointly supervised this work.
* swohl@broadinstitute.org (SW); bronwyn@broadinstitute.org (BLM)

**Data Availability Statement:** All code and data generated as part of this study are publicly available at: http://doi.org/10.5281/zenodo.3338599. The 203 mumps virus whole genome

## Abstract

Unusually large outbreaks of mumps across the United States in 2016 and 2017 raised questions about the extent of mumps circulation and the relationship between these and prior outbreaks. We paired epidemiological data from public health investigations with analysis of mumps virus whole genome sequences from 201 infected individuals, focusing on Massachusetts university communities. Our analysis suggests continuous, undetected circulation of mumps locally and nationally, including multiple independent introductions into Massachusetts and into individual communities. Despite the presence of these multiple mumps virus lineages, the genomic data show that one lineage has dominated in the US since at least 2006. Widespread transmission was surprising given high vaccination rates, but we found no genetic evidence that variants arising during this outbreak contributed to vaccine escape. Viral genomic data allowed us to reconstruct mumps transmission links not evident from epidemiological data or standard single-gene

sequences generated in this study, as well as nine low quality sequences not included in the analysis, are also available on NCBI GenBank under BioProject accession PRJNA394142 (accession numbers MF965196–MF965318 and MG986380–MG986468). All associated metadata is available in linked NCBI BioSample entries, and in S1 Data.

**Funding:** Funding was provided by: NIH NIAID U19AI110818 (Broad Institute); NIH NIAID U54GM088558 (J.A.L.); Howard Hughes Medical Institute (P.C.S.); Harvard University Burke Global Health Fellowship (P.C.S.); Amazon Web Services Cloud Credits for Research (P.C.S.). The project described was supported by award number T32GM007753 from the National Institute of General Medical Sciences (E.H.B.). The funders had no role in study design, data collection and analysis, decision to publish, or preparation of the manuscript.

**Competing interests:** The authors have declared that no competing interests exist.

**Abbreviations:** BF, Bayes factor; BSSVS, Bayesian stochastic search variable selection; BU, Boston University; CDC, US Centers for Disease Control and Prevention; CI, confidence interval; Ct, cycle threshold; F, fusion protein; Harvard, Harvard University; HN, hemagglutinin-neuraminidase; HPD, highest posterior density; MA, Massachusetts; MCC, maximum clade credibility; MDPH, Massachusetts Department of Public Health; MMR, Measles-Mumps-Rubella; MuV, mumps virus; NP, nucleoprotein; PCA, principal component analysis; PCR, polymerase chain reaction; PS, path sampling; $R_E$, effective reproduction number; ROC, receiver operating characteristic; RT-qPCR, real-time quantitative polymerase chain reaction; SH, small hydrophobic; SNP, single nucleotide polymorphism; SS, stepping-stone sampling; tMRCA, time to the most recent common ancestor; UMass, University of Massachusetts Amherst.

surveillance efforts and also revealed connections between apparently unrelated mumps outbreaks.

## Introduction

An unusually large number of mumps cases were reported in the United States in 2016 and 2017, despite high rates of vaccination [1,2]. In the prevaccination era, mumps was a routine childhood disease, with over 150,000 cases reported in the US annually [1]. After the mumps vaccine was introduced in 1967, mumps incidence declined by more than 99% [1]. Case counts rose again briefly in the mid-1980s and then continued to decrease after a national outbreak of measles prompted the recommendation of 2 Measles-Mumps-Rubella (MMR) vaccine doses in 1989 [3]. In the early 2000s, only a few hundred cases of mumps were reported annually in the US [1], attesting to the success of vaccination, possibly combined with decreasing clinical suspicion. This apparently low nationwide incidence was interrupted by an outbreak of >5,000 cases in the Midwestern US in 2006 [4], followed by a period of low incidence with minor outbreaks until 2016. This recent resurgence in mumps is partially explained by waning vaccine-induced immunity [5], but the extent to which genetic changes in circulating viruses have contributed is not yet clear.

In Massachusetts, over 250 cases were reported in 2016 and over 170 in 2017, far exceeding the usual state incidence of <10 cases per year [6]. As seen in other recent outbreaks, most cases were associated with academic institutions [4] and other close-contact settings, including prisons [7] and tightly-knit ethnic and religious communities [8,9]. Mumps was reported to the Massachusetts Department of Public Health (MDPH) by 18 colleges and universities in the state, including Harvard University (Harvard), University of Massachusetts Amherst (UMass), and Boston University (BU)—the 3 institutions with the largest numbers of reported cases. Of the individuals infected, 65% had the recommended 2 doses of the MMR vaccine (S1 Table).

We used whole genome sequencing, phylogenetic analysis, and transmission reconstruction to investigate the spread of mumps at multiple geographic scales, including within a college campus, more widely in Massachusetts, and across the US. Pathogen sequence data have become an important tool for understanding the spread of infectious diseases in near real time, allowing researchers to pinpoint outbreak origins [10,11], resolve transmission patterns [12], and detect changes throughout the genome that could affect disease severity or the effectiveness of vaccines and diagnostics [13–16]. Such data have been shown to be most useful when analyzed alongside epidemiological data [12,17,18], although the field is still exploring in detail how genomics can contribute to understanding and controlling outbreaks [19]. Mumps outbreaks in 2016 and 2017 in the US, particularly those in universities, provided an opportunity to apply these ideas to the mumps virus and to further this exploration in the context of a closely monitored, largely self-contained campus setting.

## Results and discussion

We generated 203 whole mumps virus genomes from buccal swabs from patients who tested positive by polymerase chain reaction (PCR) for mumps virus (Fig 1A), of which 158 were from Massachusetts during the 2016 and 2017 outbreak (Fig 1B), with 92 from Harvard, BU, or UMass in particular. The remaining 43 genomes were from 15 other states, collected between 2014 and 2017 (Table 1). These 203 genomes come from 259 PCR-positive samples, 56 of which were excluded because they did not produce data that met our definition of a

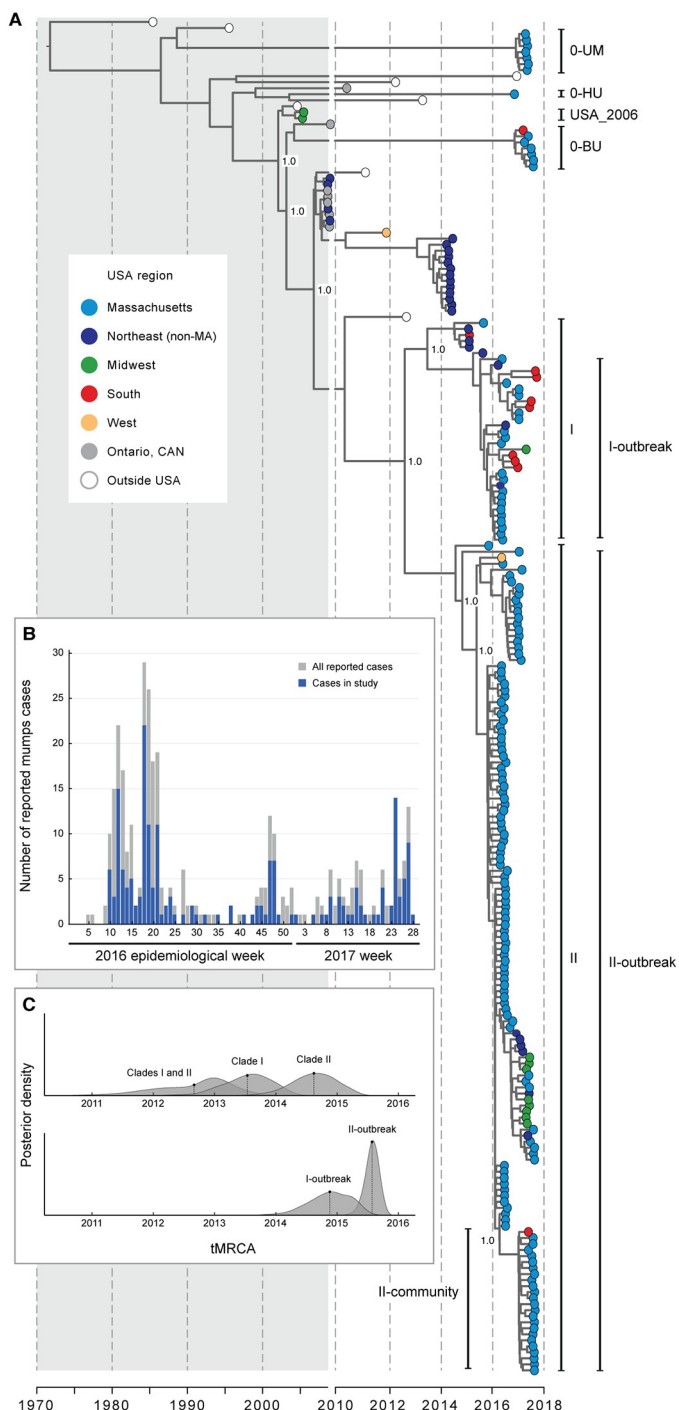

**Fig 1. Massachusetts mumps outbreak overview.** (A) Maximum clade credibility tree of 225 mumps virus genotype G whole genome sequences, including 200 generated in this study. Labels on internal nodes indicate posterior support. Clades I and II contain 91% of the samples from the 2016 and 2017 Massachusetts (MA) outbreak; I-outbreak and II-outbreak are the largest clades within them that contain only samples from the outbreak. Clade 0-UM contains samples associated with UMass other than those in Clades I and II; the same is true for 0-BU (BU) and 0-HU (Harvard). II-community contains primarily samples associated with a local Massachusetts community. (B) Number of reported mumps cases in 2016 by epidemiological week in Massachusetts (gray) and in this study (blue). (C) Probability distributions for the date of the most recent common ancestor (computed from tMRCA) of selected clades (see S2 Table for additional clades). Dotted line is the mean of each distribution. BU, Boston University; HU, Harvard Univeristy; tMRCA, time to the most recent common ancestor; UM, University of Massachusetts Amherst; UMass, University of Massachusetts Amherst.

**Table 1. Summary of samples and genomes.** Counts of samples sequenced and genomes generated by source (MDPH or CDC), date, and mumps virus PCR result. [G] indicates genotype G genomes. Two genomes are from a second sample of a patient already included in the data set.

| Source | Dates | PCR result | Samples | Genomes | Genomes (unique patients) | Genomes [G] (unique patients) |
|---|---|---|---|---|---|---|
| CDC | 2014–2015 | + | 26 | 18 | 18 | 18 |
| CDC | 2016–2017 | + | 33 | 25 | 25 | 25 |
| MDPH | 2014–2015 | + | 6 | 2 | 2 | 2 |
| MDPH | 2016–2017 | + | 194 | 158 | 156 | 155 |
| MDPH | 2016–2017 | – | 29 | 0 | 0 | 0 |
| | | Total | 288 | 203 | 201 | 200 |

**Abbreviations**: CDC, US Centers for Disease Control and Prevention; MDPH, Massachusetts Department of Public Health; PCR, polymerase chain reaction

complete genome (see Materials and methods) because of low viral loads, often caused by collecting the samples too late in the course of infection (S1A and S1B Fig). These 56 samples that did not produce genomes were well distributed across the 2016 through 2017 study period and showed no geographic clustering (Massachusetts versus elsewhere).

The median sequencing depth of the 203 successful genomes was 176x (first quintile: 42.4x; fourth quintile: 651.8x; S1C Fig) and all were >82% complete (182 genomes were >99% complete). These full genomes provide substantially more data than sequences of the small hydrophobic (SH) gene alone (accounting for <3% of the genome), which is conventionally used for classifying mumps virus [20,21]. For the Massachusetts samples, epidemiological data (including university affiliation and vaccination status) and contact tracing data collected by the MDPH and local universities were also available (S1 Table; see S1 Data for full line list). We note that 2 of the individuals in the Massachusetts data set had paired samples from 2 timepoints; in both cases, we excluded the genome from the later sample in subsequent analyses, leaving 201 genomes from unique patients. Of these 201 individuals from Massachusetts, 72% had known vaccination status, of whom 93% had 1 or more MMR doses. We also sequenced 29 PCR-negative samples from suspected mumps cases; in only one was there limited evidence for mumps virus, but we did identify 4 other viruses (one in each of 4 of these samples; S3 Table), 2 of which are known to cause parotitis [22], a characteristic symptom of mumps [23].

The viral whole genome data led to 2 key findings about the origin and spread of the recent outbreaks. First, our analysis revealed that the outbreaks—in Massachusetts and also more broadly in the US—were largely the product of a single mumps lineage and that this lineage was responsible for most if not all US mumps outbreaks since at least 2006. This lineage belongs to mumps virus genotype G [20,21] as do all but 1 of the 201 genomes from unique patients in our data set. Unless otherwise stated, all subsequent analyses refer only to these 200 genotype G genomes (Table 1).

Phylogenetic analysis of these 200 genomes, along with the other 25 publicly available genotype G genomes, shows that almost all (192 of the 200 samples in our data set; 211 of the 225 total) belong to a single lineage within genotype G, despite having been collected from all over the country (Fig 1A; also supported by a principal component analysis; see S2 Fig). This lineage descends from the US mumps outbreak in 2006. Single-gene data (see discussion of SH gene below) show no evidence for extensive transmission of this lineage outside the US, suggesting that most mumps cases in the US since 2006 are the result of ongoing transmission of a single lineage within the United States; specifically, this connects previously unassociated cases (with published genomes) between 2006 and 2016 [1,8,24]. This also suggests that unreported infections may be common and may be driving ongoing transmission [23,25].

The distribution of cases within the phylogenetic tree further suggests geographic movement of mumps virus on short time scales. For example, clade I (Fig 1A) contains virus genomes collected in 2015 through 2017 from Massachusetts, from elsewhere in the Northeast, from the South, and from the Midwest. The date of the most recent common ancestor of this clade was most likely in 2013 or later (Fig 1C and S3 Table), which implies that the virus spread to all of those regions in less than 5 years. Similarly, clade II contains genomes from all of those regions as well as one from the western US, all collected during the same time period. Clade II likely dates from no earlier than 2014, suggesting spread to these regions occurred in an even shorter amount of time. Together, these observations provide strong evidence for regional circulation in the Northeast, coupled with at least occasional movement to more distant geographic regions. Further sampling of these other regions may reveal additional long-distance dispersal events, which would indicate widespread geographic movement of mumps. This would not be surprising if transmission frequently occurs through college students, many of whom reside in close-contact settings and also travel long distances to return home.

The second key finding about the recent spread of mumps virus relates to its origins in Massachusetts. Although this 2006 lineage dominated the Massachusetts outbreak, the outbreak was not caused by a single introduction of mumps virus into the state but rather is comprised of at least 6 distinct viral clades (Fig 1A). Several of these were sublineages of the dominant 2006 lineage, including the 2 largest, Clades I and II, which comprise more than 90% of samples from Massachusetts (13% and 77%, respectively). These 2 clades diverged before the 2016 outbreak began (Fig 1C and S2 Table), indicating that they independently contributed to it. The remaining 4 clades (0-UM, 0-HU, 0-BU, plus one genotype K genome) likewise represent independent introductions, one of which (0-BU) also falls within the 2006 lineage. Such multiple introductions, indicating widespread transmission of mumps virus, have been observed elsewhere in the US [25] and are also seen in this data set within individual universities: samples from UMass span 2 clades (II and 0-UM), Harvard samples span 3 clades (I, II, and 0-HU), and BU samples span 3 clades (I, II, 0-BU) (S3 Fig). This emerging pattern of complexity, which has only become apparent with viral genomic data, provides a previously inaccessible look into the degree to which mumps is moving within and between states and communities in the US.

Epidemiological investigation showed that each of the 4 minor clades described above contains at least one sample from a patient with foreign travel history during the incubation period of the virus (12–25 days before symptom onset [26]; S1 Data). Together, the genomic and epidemiological data suggest that most mumps virus cases in the US belong to the primary 2006 lineage but that small clusters of cases can be attributed to repeated importation of the virus from outside the country. These imported cases do not appear to be important contributors to the overall burden of mumps in the US.

The finding that a single mumps lineage has been successful in a highly vaccinated population [2], despite repeated introductions of other lineages, raised the possibility that mutations within this lineage have contributed to its success, perhaps by enabling vaccine escape. Because mutations contributing to the success of the entire lineage necessarily occurred early in the evolution of that lineage, we note here the differences between fixed mutations in our data set and the strain used in the mumps vaccine.

We found that there were numerous fixed differences between samples from the 2016 outbreak and the Jeryl Lynn vaccine strain in regions of immunological significance (S4A Fig and S1 Text), consistent with a recent similar analysis [27]. In the hemagglutinin-neuraminidase (HN) protein, the primary target of neutralizing antibodies [28], we observed 32 sites with fixed amino acid substitutions between our sequences and the Jeryl Lynn strain. Thirty of

these sites were conserved between our sequences and a cell-passaged clinical strain that was isolated from Iowa in 2006 (accession: JX287385), near the beginning of mumps resurgence in the US. The Iowa 2006 strain was previously shown to be neutralized by sera from both vaccinated and naturally infected individuals, but to a lower degree than neutralization of the Jeryl Lynn strain itself [29–31], raising the possibility that some of these mutations may confer partly reduced neutralization susceptibility. In addition, we observed 2 positions at which our sequences differed from both the Jeryl Lynn and the Iowa 2006 strains. At these 2 positions, the variant observed in our sequences was also present in most other genotype G sequences published to date, including in sequences from a recent study from the Netherlands [28]. Further studies are warranted to test the neutralization susceptibility of strains containing these variants, because the Iowa 2006 sequence may not be fully representative of most currently circulating genotype G viruses.

We also looked for any evidence of ongoing adaptation to the vaccine during the outbreak. We considered this as a possibility because the vaccine was introduced relatively recently in the history of the mumps virus, recently enough that the virus could still be adapting to it. Additionally, in the absence of widespread natural infection, vaccination now constitutes the largest immunological selective force on mumps virus in the US. To investigate this, we paired genomic data with vaccination records to look for any evidence of changes in the mumps virus genome during the outbreak that led to antigenic variation from the Jeryl Lynn vaccine strain.

We first tested whether nucleotide substitutions in genomes from the Massachusetts outbreak clustered by time since vaccination, or whether vaccinated individuals clustered on certain branches of the phylogenetic tree; neither was the case (S4B and S4E Fig). Second, we looked for signals of positive selection (using the d$N$/d$S$ statistic) in the 225 genotype G genomes in our data set; a signal here would suggest that nonsynonymous mutations in a particular gene were being favored by ongoing selective pressure. We found no evidence for selection in any gene or at any specific site (S4C and S4D Fig). Thus, we did not find direct evidence of genetic variants arising within this outbreak that contributed to vaccine escape, although we note that both tests have quite limited statistical power in this data set. This finding is consistent with a recent study that proposed waning vaccine-induced immunity as a driving factor in recent US mumps outbreaks [5]; this hypothesis is also supported by our own data, in which we find that time since vaccination differs between Massachusetts individuals testing positive and negative for mumps virus by PCR in 2016 through 2017, with longer times since vaccination observed in mumps-positive patients (S5 Fig).

Understanding transmission routes can be crucial in guiding the public health response to an outbreak—for example, whether efforts should be directed toward controlling mumps spread within a university or preventing virus importation. In the Massachusetts mumps outbreak, detailed genomic data allowed us both to confirm connections suggested by public health investigation and to identify new links between cases. The phylogeny described above shows that mumps samples from different Massachusetts universities were genetically similar and fell within the 2 primary clades (S3 Fig), consistent with the epidemiological interpretation that these contemporaneous cases were part of 1 large mumps outbreak in 2016 through 2017. It also showed an unexpected connection between mumps cases in a local, nonacademic community (Clade II-community) and those at Harvard: the II-community cases fall within the predominantly Harvard Clade II, suggesting a spillover event from the university into the wider community (Fig 2A). Cases in these 2 clades were classified as distinct outbreaks during initial public health investigation based on epidemiological data [32], different demographic makeup of the 2 populations (older adults with no obvious university connection versus mostly college-aged students), and a 5-month gap between the last confirmed cases at Harvard and the cases in the local community. However, the genomic data clearly suggests a

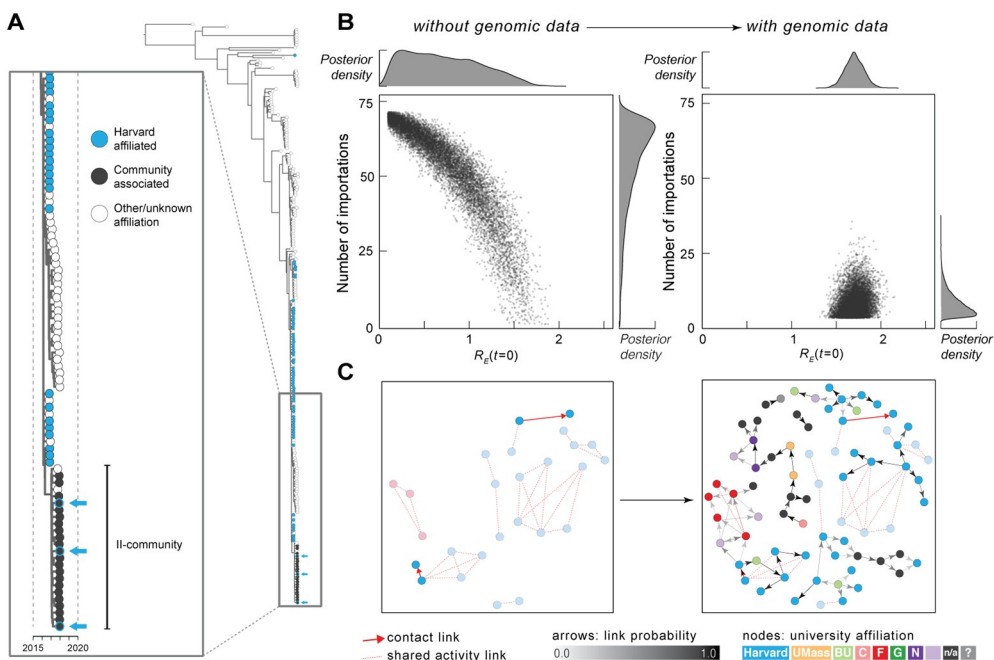

**Fig 2. Epidemiological modeling and transmission reconstruction.** (A) Zoom view of Clade II-community and its ancestors (see Fig 1A). Arrows: individuals affiliated with both II-community and Harvard. (B) Number of importations into Harvard calculated without (left) and with (right) viral genetic information as input. Each point represents a sample from the posterior distribution of $R_E(t = 0)$ and the number of introductions, based on simulated transmission dynamics. (C) Transmission reconstruction of individuals within Clade II-outbreak; samples are colored by institution affiliation (light purple: other institution; n/a: no affiliation; question mark: unknown affiliation). Left: reconstruction using epidemiological data only; all individuals in Clade II-outbreak with known epidemiological links (red arrows) are shown. Right: reconstruction using mumps genomes and collection dates. Arrow shading indicates probability of direct transmission between individuals (minimum probability shown: 0.3); cases with 1 or more inferred links are shown and are colored by institution. Arrows outlined in red represent transmission events identified by both genomic and epidemiological data. Faded nodes are those only connected by shared activity links (i.e., no inferred or known direct transmission). BU, Boston University; Harvard, Harvard University; $R_E$, effective reproduction number; UMass, University of Massachusetts Amherst.

connection, supported by additional epidemiological investigation that identified 3 individuals affiliated with both Harvard and the local community who could have acted as transmission links.

Because of the large number of cases reported and sequenced and the contact tracing information available, we were able to quantify mumps transmission dynamics within Harvard. We first used an epidemiological model (S6 Fig) [5] without genomic data to estimate transmission within the university, but it did not permit us to distinguish between a single mumps introduction followed by high transmission and multiple introductions followed by low transmission (Fig 2B left). We then modified the model to incorporate the number of viral lineages (Clades 0-HU, II-community, and 2 subclades within Clade II) observed within Harvard. This markedly improved our ability to distinguish these scenarios, supporting an estimate of 5 (95% CI: 4–18) distinct introductions to Harvard and an effective reproduction number ($R_E$) of 1.70 (95% CI: 1.50–1.91; Fig 2B right). An $R_E$ well above 1.0 means that the outbreak could be self-sustaining, which highlights the importance of controlling on-campus transmission during mumps outbreaks.

The high-resolution data from Harvard allowed us to estimate transmission links between individual cases, which can aid in targeting containment efforts aimed at high-risk individuals

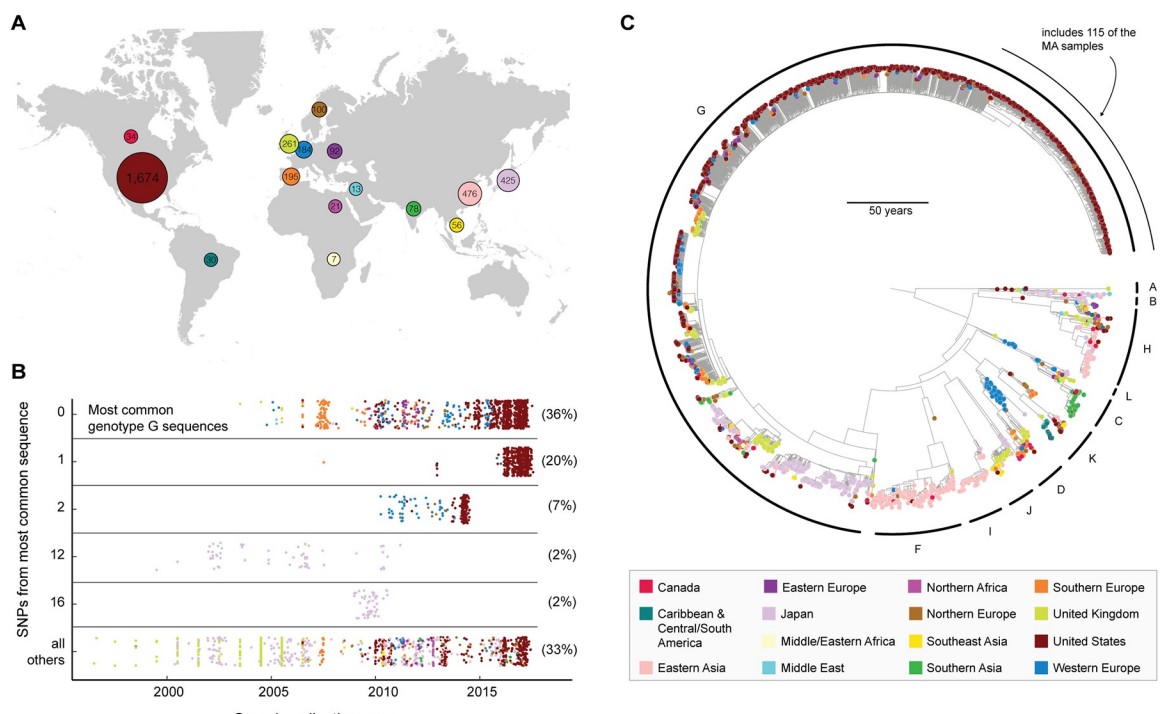

**Fig 3. Global spread of mumps virus based on SH gene sequences.** Colors in all panels are by region (legend in bottom right). (A) Number of SH sequences in our data set from each of the 15 regions. (B) Identical genotype G sequences over time from 1995 through 2017. Each dot represents a sample; each row contains samples with identical SH sequences, except the bottom, which includes samples with sequences distinct from those in the above 5 categories. Numbers on the right: percentage of all genotype G samples found in that row. (C) Maximum clade credibility tree of 3,646 publicly available SH gene sequences, including 193 complete SH sequences generated in this study. SH, small hydrophobic.

or groups. For this purpose, we focused on Clade II-outbreak because it was largely contained within a single institution (Harvard) and had dense sampling. When we attempted to link individual mumps cases within this clade using contact tracing data alone, we could only infer direct mumps transmissions ("contact links") between 2 pairs of individuals (Fig 2C left). We then used the genomic data to validate genetic distance as a proxy for epidemiological linkage (S7A and S7B Fig) and, based on these results, used genomic data and sampling dates to reconstruct possible mumps transmissions. The reconstruction (Fig 2C right, see also S7C Fig), which estimates and assigns a probability to direct transmission links, supports 1 of the 2 pairs of direct mumps transmissions as well as 2 "shared activity links" (see Materials and methods) and also suggests many new links between individuals without any known contacts. Indirect links, however, are difficult to identify using genomic data alone, illustrating the complementary contributions of genomic and epidemiological data for reconstructing detailed transmission, which can guide response efforts for mumps and other outbreaks in close-contact settings.

Conventional sequence-based mumps surveillance has been limited to the SH gene. The SH gene is a small (316-nucleotide), convenient target for sequencing [20,33] and is thus the region for which the most sequence data are available. We used the 3,646 publicly available SH sequences from mumps cases around the world (Fig 3A) to assess whether SH sequences would be adequate to distinguish the lineages and transmission patterns identified above. To do this, we constructed the mumps phylogeny using the SH sequence alone and compared it to the whole genome phylogeny. There was limited variation within each genotype (genotype

G shown in Fig 3B), and neither of our main phylogenetic conclusions based on whole genome data could be ascertained from SH alone: we could not determine the relationship between the spillover into the local community and its source in Harvard (other than that they belonged to the same clade), and we were unable to determine the relationship between the 2016 and 2017 outbreak and the 2006 lineage (S8 Fig). Our other key findings, such as the detailed picture of transmission within Harvard and a refined $R_E$ estimate, rely on these 2 conclusions. Nevertheless, SH data were sufficient to resolve population structure on a global scale and to confirm that the 11 known clinically relevant genotypes are associated with particular world regions (Fig 3C) [34]. Moreover, an analysis of global migration reveals significant movement of mumps virus between the US and Europe. For details on the global SH analysis, including a discussion of its limitations, see S1 Text, S9 and S10 Figs.

The combination of high-quality genomic and epidemiological data from the Massachusetts mumps outbreak revealed the extent to which mumps is circulating in the US, connected previously unrelated outbreaks, and allowed us to trace transmission within and between individual communities. Given the high-quality genomic data we were able to produce from mumps clinical samples, as well as the limited information that can be gleaned from SH sequencing, it is worth considering whether future public health surveillance of mumps should incorporate whole genome sequencing. The collection of these detailed data, which we have made available to the community (see Materials and methods), was only possible through extensive collaboration between state and national public health agencies, academic researchers, and affected universities throughout the greater Boston, Massachusetts area. We hope that these partnerships, fostered in response to a surge in mumps cases in Massachusetts in 2016 and 2017, will facilitate real-time genomic and epidemiological data generation, analysis, and sharing in future outbreaks of any pathogen.

## Materials and methods

### Ethics statement

The study protocol was approved by the MDPH, Centers for Disease Control and Prevention (CDC), and Massachusetts Institute of Technology (MIT) Institutional Review Boards (IRB) (MDPH IRB 00000701, project 906066). Harvard University Faculty of Arts and Sciences and the Broad Institute ceded review of sequencing and secondary analysis to the MDPH IRB through authorization agreements. The MDPH IRB waived informed consent given this research met the requirements pursuant to 45 CFR 46.116 (d). The CDC IRB determined this project to be nonhuman subjects research as only deidentified leftover diagnostic samples were utilized. In compliance with the IRB agreement, Harvard University, University of Massachusetts Amherst, and Boston University granted approval for publication of their institution names in this paper.

### Sample collections and study subjects

Buccal swab samples were obtained from suspected and confirmed mumps cases tested at MDPH and CDC. Samples from MDPH ("Cases in Study," Fig 1B) include all cases with a positive mumps PCR result (see "PCR diagnostic assays performed at MDPH and CDC" below) collected between 1 January 2014 and 30 June 2017. Demographic information for all cases reported in Massachusetts (Fig 1B and Table 1) includes all confirmed and probable mumps cases reported to MDPH in that time period. Probable cases include cases with a positive mumps IgM assay result or those with an epidemiological link to a confirmed case [35]. Samples from CDC are a selection of PCR-positive cases submitted to the CDC for testing between

2014 and 2017. See S1 Data for deidentified information, including metadata, about study participants.

## Viral RNA isolation

Sample inactivation and RNA extraction were performed at the MDPH, Broad Institute, and CDC. At MDPH, viral samples were inactivated by adding 300 μL Lysis/Binding Buffer (Roche) to 200 μL sample, vortexing for 15 seconds, and incubating lysate at room temperature for 30 minutes. RNA was then extracted following the standard external lysis extraction protocol from the MagNA Pure LC Total Nucleic Acid Isolation Kit (Roche) using a final elution volume of 60 μL. At the Broad Institute, samples were inactivated by adding 252 μL Lysis/Binding Buffer (ThermoFisher) to 100 μL sample. RNA was then extracted following the standard protocol from the MagMAX Pathogen RNA/DNA Kit (ThermoFisher) using a final elution volume of 75 μL. At CDC, RNA extraction followed the standard protocol from the QiaAmp Viral RNA mini kit (Qiagen).

## PCR diagnostic assays performed at MDPH and CDC

Diagnostic tests for presence of mumps virus were performed at the MDPH and CDC using the CDC Real-Time (TaqMan) RT-PCR Assay for the Detection of Mumps Virus RNA in Clinical Samples [8,36]. Each sample was run in triplicate using both the Mumps N Gene assay (MuN) and RNase P (RP) assay using this protocol. RT-PCR was performed on the Applied Biosystems 7500 Fast Real-Time PCR system or Applied Biosystems Prism 7900HT Sequence Detection System instrument.

## PCR quantification assays performed at Broad Institute

Mumps virus RNA was quantified at the Broad Institute using the Power SYBR Green RNA-to-Ct 1-Step qRT-PCR assay (Life Technologies) and CDC MuN primers. The 10 μL assay mix included 3 μL RNA, 0.3 μL each of mumps virus forward and reverse primers at 5 μM concentration, 5 μL 2x Power SYBR RT-PCR Mix, and 0.08 μL 125x RT Enzyme Mix. The cycling conditions were 48 ˚C for 30 minutes and 95 ˚C for 10 minutes, followed by 45 cycles of 95 ˚C for 15 seconds and 60 ˚C for 30 seconds with a melt curve of 95 ˚C for 15 seconds, 55 ˚C for 15 seconds, and 95 ˚C for 15 seconds. RT-PCR was performed on the ThermoFischer QuantStudio 6 instrument. To determine viral copy number, a double-stranded gene fragment (IDT gBlock) was used as a standard. This standard is a 171 bp fragment of the mumps genome (GenBank accession: NC_002200) including the amplicon (sequence: GGA TCG ATG CTA CAG TGT ACT AAT CCA GGC TTG GGT GAT GGT CTG TAA ATG TAT GAC AGC GTA CGA CCA ACC TGC TGG ATC TGC TGA TCG GCG ATT TGC GAA ATA CCA GCA GCA AGG TCG CCT GGA AGC AAG ATA CAT GCT GCA GCC AGA AGC CCA AAG GTT GAT TCA AAC).

23S rRNA content in samples was quantified using the same Power SYBR Green RNA-to-Ct 1-Step qRT-PCR assay kit and cycling conditions. Primers were used to amplify a 183 bp universally conserved region of the 23S rRNA (fwd: 93a - GGG TTC AGA ACG TCG TGA GA, rev: 97ar—CCC GCT TAG ATG CTT TCA GC) [37]. To determine viral copy number, a double-stranded gene fragment (IDT gBlock) was used as a standard. This standard is a 214 bp fragment of the Streptococcus HTS2 genome (accession: NZ_CP016953) (sequence: AGC GGC ACG CGA GCT GGG TTC AGA ACG TCG TGA GAC AGT TCG GTC CCT ATC CGT CGC GGG CGT AGG AAA TTT GAG AGG ATC TGC TCC TAG TAC GAG AGG ACC AGA GTG GAC TTA CCG CTG GTG TAC CAG TTG TCT CGC CAG AGG CAT CGC TGG GTA GCT ATG TAG GGA AGG GAT AAA CGC TGA AAG CAT CTA AGT

GTG AAA CCC ACC TCA AGA T). Data from both assays—each performed only on a subset of samples—is reported in S1 Data.

## Bacterial rRNA depletion

Bacterial rRNA was depleted from some RNA samples (see S1 Data) using the Ribo-Zero Bacteria Kit (Illumina). At the hybridization step, the 40 μL reaction mix included 5 μL RNA sample, 4 μL Ribo-Zero Reaction Buffer, 8 μL Ribo-Zero Removal Solution, 22.5 μL water, and 0.5 μL synthetic RNA (25 fg) used to track potential cross-contamination (gift from M. Salit, NIST). Bacterial rRNA-depleted samples were purified using 1.8x volumes Agencourt RNA-Clean XP beads (Beckman Coulter) and eluted in 10 μL water for cDNA synthesis.

## Illumina library construction and sequencing

cDNA synthesis was performed as described in previously published RNA-seq methods [38]. In samples in which bacterial rRNA was not depleted, 25 fg synthetic RNA was added at the beginning of cDNA synthesis to track sample cross-contamination. Positive control libraries were prepared from a mock mumps virus sample in which cultured Enders strain (ATCC VR-106) mumps was spiked into a composite buccal swab sample from healthy patients and diluted to mumps virus RT-qPCR Ct = 21. This mock sample was extracted using the viral RNA isolation protocol described above, except that total nucleic acid was eluted in 100 μL. Negative control libraries were prepared from nuclease-free water. Illumina Nextera XT was used for library preparation: indexed libraries were generated using 16 cycles of PCR, and each sample was indexed with a unique barcode. Libraries were pooled equally based on molar concentration and sequenced on the Illumina HiSeq 2500 (100 or 150 bp paired-end reads) platform.

## Hybrid capture

Viral hybrid capture was performed as previously described [38] using 2 different probe sets. In one case, probes were created to target mumps and measles virus (V-MM probe set), and in one case, probes were created to target 356 species of viruses known to infect humans (V-All probe set) [39]. Capture using V-All was used to enrich viral sequences primarily in samples in which we could not detect mumps virus, as well as in other samples (see S1 Data for a list of which samples were captured using which probe set). As described in the work by Metsky and colleagues [39], the probe sets were designed to capture the diversity across all publicly available genome sequences on GenBank for these viruses. Probe sequences can be downloaded here: https://github.com/broadinstitute/catch/tree/cf500c69/probe-designs.

## Genome assembly

We used viral-ngs version 1.18.1 [40] to assemble genomes from all sequencing runs. Viral-ngs is freely available under a BSD license (https://viral-ngs.readthedocs.io/en/latest/). We used a set of mumps sequences (accessions: JX287389.1, FJ211586.1, AB000386.1, JF727652.1, AY685920.1, AB470486.1, GU980052.1, NC_002200.1, AF314558.1, AB823535.1, AF467767.2) to taxonomically filter these reads. We de novo assembled reads and scaffolded against the mumps genome with accession JX287389.1 to assemble a genome for each replicate. Then, we pooled read data from all sequencing replicates of each sample and repeated this assembly process to obtain final genomes. Each time we ran viral-ngs, we set the "assembly_min_length_fraction_of_reference" and "assembly_min_unambig" parameters to 0.01. Technical replicates had high concordance: in 27 samples prepared more than once, only 2 base calls differed across replicates.

We replaced deletions in the coding regions with ambiguity ("N"). In one sample, MuVs/Massachusetts.USA/11.16/5 [G], with an insertion at position 3,903 (based on a full 15,384-nucleotide mumps virus genome, e.g., accession JN012242.1) we removed a poorly supported (<5 reads covering the site) extra "A" in a homopolymer region.

To calculate sequencing metrics (S1 Fig), we used SAMtools [41] to downsample raw reads for each replicate to 1 million reads and then reran assembly as described above. Samples from 1 contaminated sequencing batch were excluded, as were all replicates from PCR-negative samples. In cases in which samples from 2 time points were sequenced from a single patient, we included only the first time point in the collection interval analysis (S1D Fig).

### Metagenomic analysis

We used the V-All probe set for capture on all samples from suspected mumps cases with a negative mumps PCR result (*n* = 29). A subset of PCR-positive samples was also sequenced with this probe set (*n* = 145; without capture = "unbiased," *n* = 111). We used the mock Enders strain mumps sample as a positive control on a sequencing run containing all PCR-negative samples, as well as a water sample as a negative control. We used the metagenomic tool Kraken version 0.10.6 [42] via viral-ngs to identify the presence of viral taxa in each sample. We built a database similar to the one described in the work by Metsky and colleagues [39], except without insect species. This database encompasses the known diversity of viruses known to infect humans. It is publicly available, in 3 parts, at https://storage.googleapis.com/sabeti-public/meta_dbs/kraken_full_20170522/ [*file*], where [*file*] is database.idx.lz4 (595 MB), database.kdb.lz4 (75 GB), and taxonomy.tar.lz4 (66 MB). Because of the possibility of contamination, we prepared a second, independent sequencing replicate on all PCR-negative samples with evidence for mumps or another virus and required both replicates to contain reads matching the virus detected in the sample. We found no evidence of pathogenic viruses other than mumps in PCR-positive samples.

We required the total raw read count for any genus in any sample to be twice (in practice, 7 times) that in any negative control from any sequencing batch. For any sample that had one or more pathogenic viral genera that passed this filter and had deduplicated reads well distributed across the relevant viral genome, we attempted contig assembly: we used viral-ngs to filter all sample reads against all NCBI GenBank [43] entries matching the identified species and then de novo assembled reads using Trinity [44] through viral-ngs and scaffolded against the closest matching full genome identified by a blastn query [45]. We report all viruses identified via this method in S3 Table.

In parallel, we used SPAdes [46] within viral-ngs to de novo assemble contiguous sequence from all de-duplicated, depleted reads. We used the metagenomic tool DIAMOND version 0.9.13 [47] with the nr database downloaded 29 May 2017, followed by blastn [45] of DIAMOND-flagged contigs. Using this method, we confirmed the presence of all previously identified viruses except influenza B virus, for which we never assembled a contiguous sequence. We found no evidence of additional pathogenic viruses using this method.

### Criteria for pooling across replicates

We prepared one or more sequencing libraries from each sample and attempted to sequence and assemble a genome from each of these replicates. We required a replicate of a sample to contain 3,000 unambiguous base calls for its read data to be included in that sample's final genome assembly. This threshold was based on the maximum number of unambiguous bases (2,820) observed in negative controls across all uncontaminated sequencing batches. One sequencing batch showed evidence of contamination: we were able to assemble 7,615

unambiguous mumps bases from a water sample, with a median coverage of 4x. For samples prepared in this batch only, we implemented an additional requirement for including a replicate in pooling: the assembly must have a median depth of coverage of $\geq$20x, 5 times the median depth of coverage of the water sample.

### Multiple sequence alignment of genotype G whole genomes

We required a mumps genome to contain 11,538 unambiguous base calls (75% of the total 15,384-nucleotide genome with GenBank accession JN012242.1) for inclusion in the alignment of whole genome sequences that we used for downstream analysis. For 2 patients with samples taken at 2 time points (MuVs/Massachusetts.USA/19.16/5 [G] (1) and MuVs/Massachusetts.USA/19.16/5 [G] (2–20.16); MuVs/Massachusetts.USA/16.16/6 [G] (1) and MuVs/Massachusetts.USA/16.16/6 [G] (2–17.16)), we only included the earlier sample in downstream analyses. The final alignment of whole genome sequences contains only samples belonging to genotype G; we did not include MuVs/Massachusetts.USA/24.17/5 [K], which belongs to genotype K, in the alignment.

In this alignment, we also included 25 mumps virus genomes published on NCBI GenBank [43]. These comprise all of the sequences with organism "Mumps rubulavirus" available as of September 2017 that meet the following criteria: sequence length $\geq$14,000 nucleotides, belong to genotype G, sample collection year and country of origin reported in GenBank, no evidence of extensive virus passaging or modification (for vaccine development, for example). The accessions are KY969482, KY996512, KY996511, KY996510, KY680540, KY680539, KY680538, KY680537, KY006858, KY006857, KY006856, KY604739, KF738114, KF738113, KF481689, KM597072, JX287391, JX287390, JX287389, JX287387, JX287385, JN012242, JN635498, AF280799, EU370207.

We aligned mumps virus genomes using MAFFT version 7.221 [48] with default parameters. We provide the sequences and alignments used in analyses at http://doi.org/10.5281/zenodo.3338599.

### Visualization of coverage depth across genomes

We plotted aggregate depth of coverage across the 200 samples whose genomes were included in the final alignment (S1C Fig) as described in the work by Metsky and colleagues [49]. We aligned reads against the reference genome with accession JX287389.1 and plotted over a 200-nt sliding window.

### Analysis of within- and between-sample variants

We ran V-Phaser 2.0 [50] via viral-ngs on all pooled reads mapping to a sample assembly to identify within-sample variants (S2 Data). To call a variant, we required a minimum of 5 forward and reverse reads, as well as no more than 10-fold strand bias, as previously described [51]. Samples with genomes generated by the sequencing batch that showed evidence of contamination (see "Criteria for pooling across replicates" above) were not included in within-host variant analysis. When analyzing variants in known contacts, we used pairs of samples designated as "contact links," as described in "Relationship between epidemiological and genetic data" below.

Between-sample variants were called by comparing each final genome sequence to JX287385.1, the earlier of the 2 available whole genomes from the 2006 mumps outbreak in Iowa, US (S2 Data). We ignored all fully or partially ambiguous base calls and excluded sequences that did not descend from the USA_2006 clade from this analysis. When examining

amino acid changes in HN given vaccination status (see "SH and HN multiple sequence alignment" below), we ignored sequences from patients with unknown vaccination history.

## Maximum likelihood estimation and root-to-tip regression

We generated a maximum likelihood tree using the whole genome genotype G multiple sequence alignment. We used IQ-TREE version 1.3.13 [52] with a GTR substitution model and rooted the tree on the oldest sequence in this data set (accession KF738113.1) in FigTree version 1.4.2 [53].

To estimate root-to-tip distance of samples in the primary US lineage, we subsetted the full genotype G alignment to include only samples descendent of the USA_2006 clade, including samples in this clade (see Fig 1A) and used TempEst version 1.5 [54] with the best fitting root (heuristic residual mean squared function) to estimate distance from the root. We used scikit-learn version 0.14.1 [55] in Python to perform linear regression of distances on dates.

We also generated maximum likelihood trees using the SH gene only (full 316-nucleotide mRNA), HN (coding region only), F (coding region only), and a concatenation of the aforementioned SH, HN, and F regions (S8 Fig). For each tree, we started with the whole genome genotype G alignment (225 sequences) and extracted the relevant region(s). We then removed any sequence with 2 or more consecutive ambiguous bases ("N"s) in any of SH, HN, or F, leaving 209 sequences in each alignment. We used IQ-TREE version 1.5.5 with a GTR substitution [56] model to generate maximum likelihood trees.

## Molecular dating using BEAST

We performed all molecular clock analyses on whole genome sequences using BEAST version 1.8.4 [57]. We excluded from the CDS the portion of the V protein after the insertion site [58] because of reading frame ambiguity in that region. On the CDS, we used the SRD06 substitution model [59], which breaks codons into 2 partitions (positions [1+2] and 3) with HKY substitution models [60] and allows gamma site heterogeneity [61] (4 categories) on each. We used a separate partition on noncoding sequence with an HKY substitution model and gamma site heterogeneity. To accommodate inexact dates in 7 sequences from NCBI GenBank, we used sampled tip dates [62].

We tested 6 models as described in the work by Metsky and colleagues [49]. Each was a combination of one of 2 clock models (strict clock and uncorrelated relaxed clock with log-normal distribution [63]) and 1 of 3 coalescent tree priors (constant size population, exponential growth population, and Bayesian Skygrid model [64] with 20 parameters). On each model, we estimated marginal likelihood with path-sampling (PS) and stepping-stone sampling (SS) [65,66] (S2 Table) after sampling 100 path steps each with a chain length of 2 million.

We sampled trees and other parameters on each model by running BEAST for 200 million MCMC steps, sampling every 20,000 steps, and removing 20 million steps as burn-in. We report the mean clock rate as the substitution rate for relaxed clock models. On the sampled trees, we used TreeAnnotator version 1.8.4 to find the maximum clade credibility (MCC) tree and visualized it in FigTree version 1.4.3 [53]. To estimate tMRCAs (Fig 1C and S2 Table), we ran BEAST again for each of the 6 models, drawing from these same sets of sampled trees (without any parameters), for 10,000 steps, sampling every step. We selected a relaxed clock and Skygrid model for plots of tMRCA distributions (Fig 1C) and MCC trees over the whole genome genotype G sequences.

Additionally, we plotted (S11 Fig) a Skygrid reconstruction of the scaled population size ($N_e\tau$) using results from the selected model. The earliest time point on this plot is the lower (more recent) 95% HPD bound on the estimated root height.

## Gene- and site-specific dN/dS analyses

We used BEAST version 1.8.4 [57] to estimate d*N*/d*S* per-site (S4C Fig and S3 Data) and per-gene (S4D Fig) using the same alignment of 225 whole genome sequences described above (again, removing the portion of the V gene after the insertion site).

For site-specific d*N*/d*S* estimation, we used the CDS as input and created a separate partition for each codon position (3 partitions). We used an HKY substitution model [60] on each partition and an uncorrelated relaxed clock with log-normal distribution [63] for branch rates. Here, we sampled from the same set of trees that were sampled as described above in "Molecular Dating using BEAST" (relaxed clock with Skygrid tree prior). We ran BEAST for 10 million MCMC steps, sampling every 10,000 steps. We estimated site-specific d*N*/d*S* at each sampled state using renaissance counting [67,68] and show summary statistics at each site after discarding 1 million steps as burn-in.

For per-gene estimation, we created 8 separate partitions: 7 correspond to the CDS of a gene (F, HN, L, M, NP, SH, partial V), and the last corresponds to noncoding sequence. For each gene partition, we used a Goldman-Yang codon model [69] with its own parameters for d*N*/d*S* (omega) and clock rate. For the noncoding partition, we used an HKY substitution model [60] and gamma site heterogeneity [61] (4 categories). We sampled tip dates as with the molecular clock analyses above and used a Bayesian Skyline tree prior [70] (10 groups). We ran BEAST for 200 million MCMC steps to sample trees and parameter values, discarded 20 million steps as burn-in, and plotted the posterior distribution of omega for each gene partition.

## Principal component analysis

The data set for PCA consisted of all SNPs from sites with exactly 2 alleles in the set of all genotype G genomes. We imputed missing data with the R package missMDA [71] and calculated principal components with the R package FactoMineR [72]. We discarded 14 samples as outliers based on visual inspection, leaving 211 samples in the final set.

## Relationship between epidemiological and genetic data

We obtained detailed epidemiological data for samples shared by MDPH from the Massachusetts Virtual Epidemiologic Network (MAVEN) surveillance system, an integrated web-based disease surveillance and case management system [73]. We defined 2 types of epidemiological links: "contact links," between individuals who were determined to be close contacts during public health investigation and had symptom onset dates 7 to 33 days apart (individuals with mumps are usually considered infectious 2 days before through 5 days after onset of parotid swelling, with a typical incubation period of 16–18 days, ranging from 12–25 days) [26]; and "shared activity links," between individuals who participated in the same extracurricular activity (e.g., a sports team or university club) or frequented a specific residence or athletic facility. When we refer to epidemiological links without specifying link type, we include both types of links.

We calculated pairwise genetic distance between all pairs of samples in the whole genome genotype G alignment. For each pair, the genetic distance score is *s*/*n*, in which *s* is the number of unambiguous differing sites (both sequences must have an unambiguous base at the site, and the called bases must differ) and *n* is the number of sites at which both sequences have an unambiguous base call.

To visualize the similarity between genomes and its relationship to epidemiological linkage, we performed a multidimensional scaling on sequences in Clade II-outbreak (Fig 1A). This clade is comprised of mostly cases from Harvard and the related community outbreak. Using their pairwise genetic distances, we calculated a metric multidimensional scaling to 2

dimensions in R with cmdscale [74]. We then evenly split the range of the output coordinates into a 100 × 100 grid and collapsed each point into this grid and plotted the number of points at each grid coordinate; this improves visualization of nearly overlapping points (identical or near-identical genomes). We plotted curves that represent epidemiological links between cases within each of the grid coordinates. This is shown in S7A Fig.

To determine the ability of genetic distance to predict epidemiological linkage, we again looked specifically at cases within Clade II-outbreak (Fig 1A). Using the Python scikit-learn package [55], we constructed a receiver operating characteristic (ROC) curve using pairwise distance between II-outbreak cases as the predictor variable and presence or absence of an epidemiological link as the binary response variable. This is shown in S7B Fig.

## Model of mumps transmission in a university setting

We developed a stochastic model for mumps virus transmission accounting for the natural history of infection, vaccination status, and control measures implemented in response to the outbreak at Harvard. Our stochastic model of mumps virus transmission included the stages after initial infection, the durations of which we inferred using data from previous clinical studies (S6A and S6C Fig). These included the gamma-distributed incubation period from infection to onset of mumps virus shedding in saliva [75]; the gamma-distributed period of latent infection from shedding onset to parotitis onset [75,76]; and the log-normally distributed time from parotitis onset to the cessation of shedding (defined in the work by Polgreen and colleagues [77]). For asymptomatic cases, we defined the total duration of shedding ($\gamma$) as the sum of independent random draws from the durations of shedding before and after parotitis onset, based on the lack of any reported difference in durations of shedding for symptomatic and asymptomatic cases [75]. To account for case isolation interventions implemented at Harvard, we modeled the removal of symptomatic individuals one day after onset of parotitis. In comparison to the 70% probability for symptoms given infection among unvaccinated individuals [78], we modeled the probability of symptoms given infection as uniformly distributed between 27.3% and 38.3% [5,79].

We used previous estimates of the effectiveness and waning rate of mumps vaccination [5] and of the vaccination status distribution of individuals on a university campus [80] to account for susceptibility to infection among the Harvard population ($N$ = 22,000). We scaled risk for mumps infection, given exposure, to time since receipt of the last vaccine dose, yielding the hazard ratio

$$\xi_i = e^{\omega_0} \tau_i^{\omega_1}$$

for an individual $i$ who received their last dose $\tau_i$ years previously, relative to an unvaccinated individual. For fitted values from the work by Lewnard and Grad [5], estimates were below 1.0 for individuals vaccinated since 1967, when the Jeryl Lynn vaccine was introduced (S6D Fig).

Given the instantaneous hazard of infection for an as yet uninfected individual $i$ exposed to $I(t)$ infected individuals

$$\lambda_i(t) = \beta \xi_i I(t) N^{-1},$$

the probability of evading infection over the course of a 1-day simulated time step was $exp$ $(-\lambda_i(t))$. The per-contact transmission rate ($\beta$) was measured from the initial (preintroduction) value of the effective reproductive number:

$$\beta = R_E(0) \bar{\gamma} \xi^{-1}.$$

## Inferring transmission dynamics

The number of cases (71) and identification of multiple, distinct viral clades within Harvard suggested limited permeation of mumps after any introduction. We simulated dynamics of individual transmission chains to understand the epidemiological course of introduced viral lineages and to infer values of $R_E(0)$ and the number of importations of mumps virus. We used the simulation model to sample from the distribution of the number of cases ($X$, including the index infection if symptomatic) resulting from a single introduction over a 1.5-year time course:

$$f\{x_i \mid R_E(0)\} = P[X = x_i \mid R_E(0)].$$

We resampled according to $f\{x_i | R_E(0)\}$ to define the distribution of the cumulative number of cases ($Z$) resulting from $Y$ introductions, conditioned on $R_E(0)$:

$$g\{z_k \mid R_E(0), Y\} = P[Z = z_k = \sum_{i=1}^{y_j} x_i \mid R_E(0), Y = y_j].$$

Of the 71 cases at Harvard, 66 had mumps genomes in our data set, so we ran simulations where $Z \geq 66$, drawing $k = 66$ cases at random to determine the number of distinct lineages ($S$, defined by the index infection) expected to be present within such a sample. The probability of obtaining 66 sequences and observing $S = s_m$ lineages among them is

$$h\{s_m \mid R_E(0), Y, K = 66\} = P[S = s_m \mid R_E(0), Y = y_j, K = 66] \times P[Z \geq 66 \mid R_E(0), Y].$$

The posterior density of our model also accounted for the probability of observing 71 symptomatic cases in total. Defined in terms of the number of introductions and the initial reproductive number, the model posterior was proportional to

$$h\{4 \mid R_E(0), Y, K = 66\} \times g\{71 \mid R_E(0), Y\},$$

where 4 is the number of viral lineages in the 66 Harvard cases (representing clades 0-HU, II-community, and 2 subclades within Clade II). We measured this probability from 100,000 iterates for each pairing of $R_E(0) \in \{0.10, 0.11, \ldots, 2.50\}$ and $Y \in \{1, 2, \ldots, 200\}$.

Last, we defined the minimum necessary third-dose vaccine coverage ($C$) to bring the effective number below unity using the relation

$$[1 - VE(0)] \times CR_E(t = 0) \leq 1$$

sampling from previous estimates [5] that mumps vaccination protects against infection, prior to waning of immunity (here defined as $VE(0)$).

## Transmission reconstruction using outbreaker

We used the R package outbreaker version 1.1–7 [81] to reconstruct transmission for samples included in Clade II-outbreak. We estimated the generation interval by fitting a gamma distribution, via maximum likelihood, to the time between symptom onset dates for cases with confirmed epidemiological links (S6E Fig). We used the same distribution for the colonization time and set the maximum number of generations between a case and its most recent sampled ancestor to 40. The resulting estimates are nearly identical to those reported in previous studies [82]. We ran outbreaker 6 times in parallel, each with 1 million MCMC steps, and discarded the first 10% of states as burn-in. We assessed run convergence and combined results for 5 of the 6 parallel runs to determine the reconstructed transmission tree (Fig 2C right). For each link in the reconstruction, the support is the frequency of the link in the samples from the

posterior (excluding the burn-in). To reconstruct transmission using SH sequences only (S7C Fig), we extracted the SH gene from the II-outbreak alignment and ran outbreaker as described above, using the results from all 6 parallel runs in the analysis.

## SH and HN multiple sequence alignment

To analyze all published SH and HN mumps sequences, we searched NBCI GenBank in July 2017 for all nucleotide sequences with organism "Mumps rubulavirus." We performed a pairwise alignment between each sequence $s_i$ and a reference genome (accession: JX287389.1) using MAFFT version 7.221 [48] with parameters: "—localpair—maxiterate 1000—preserve-case." We then extracted the SH sequence from each $s_i$ based on the reference coordinates in the alignment, removing all SH sequences without the full 316-nucleotide region and all SH sequences with an insertion or deletion ("indel") relative to the reference. We then used MAFFT with parameters "—localpair—maxiterate 1000—retree 2 –preservecase" to create a multiple sequence alignment of the extracted SH gene sequences and removed any sequences with indels in this final alignment. We repeated the same process for the HN region, requiring the full 1,749-nucleotide coding region.

In both the SH and HN alignments, we removed sequences from vaccine strains (i.e., genotype N, or another genotype marked as "(VAC)" or "vaccine"). We also removed sequences with GenBank records indicating extensive passaging. In the SH alignment only, we removed sequences with no reported collection date or country of origin, because these data are required for phylogeographic analyses. In samples with a collection decade (e.g., 1970s) but not a specific year, we assigned the first year of the decade; in samples with only a collection year, we assigned a decimal year of $year + 0.5$ (e.g., 1970.5); in samples with year and month but no day, we used the day halfway through the given month (e.g., 2015–03 becomes 2015-03-15) to calculate the decimal year; and in samples with an epidemiological week but no specific day, we approximated the decimal year as $year + (epi\ week\ /\ 52)$, except samples collected in epidemiological week 52 were relabeled as week 51.999 to avoid confusion with year-only samples.

In both the SH and HN alignments, we relabeled outdated genotypes (M, E, and any subgenotypes [21]) and constructed a maximum likelihood tree (using IQ-TREE with a GTR substitution model, as described above) to assign a genotype if one was not reported on GenBank. We preserved genotypes designated as "Unclassified" [21].

To each alignment, we added all SH or HN sequences from individual patients generated in this study, except those with 2 or more consecutive ambiguous bases ("N"s) in the SH or HN region. The sequences used in the SH and HN analyses are listed in S4 Data.

## SH phylogeographic analysis

To perform phylogenetic and phylogeographic analyses of the SH gene sequence, we first sampled trees using BEAST version 1.8.4 [57]. We used constant size population and strict clock models and used the HKY substitution model [60] with 4 rate categories and no codon partitioning. We ran BEAST in 4 replicates, each for 500 million states with sampling every 50,000 states, and removed the first 150 million states as burn-in. We verified convergence of all parameters across the 4 replicates and then combined the 4 replicates using LogCombiner.

We used TreeAnnotator to determine the MCC tree (Fig 3C) from a resampling of 350 trees and visualized the result in FigTree version 1.4.3 [53]. We computed a kernel density estimate of the probability distribution of the tMRCA over all sampled states for the 11 genotypes in this data set (S9 Fig).

To construct distributions of estimates, we used resampling on the input sequences, similar to prior work facing sampling biases [83]. To perform this resampling, we focused on only samples that were collected both within a window of time and from a geographic region with sufficient sampling. Namely, we considered only sequences sampled in 2010 or afterward and collapsed the locations shown on the full data set (Fig 3A) to just 4 global regions: US (consisting of only samples from the US), Europe (consisting of samples whose location was labeled as Eastern Europe, Northern Europe, Southern Europe, Western Europe, or the United Kingdom), East Asia (consisting of samples whose location was labeled as Eastern Asia or Japan), and South/Southeast Asia (consisting of samples whose location was labeled as Southeast Asia or Southern Asia). These 4 regions encompass 3,541 of the 3,646 SH gene sequences used in our analysis. We ignored samples from 5 locations: Canada; Caribbean, Central America, and South America; Middle and Eastern Africa; Northern Africa; Middle East. See S10A Fig for a visual representation of these 4 regions. Then, we randomly sampled 10 sequences (without replacement) from each region for each year (i.e., 2010–2011, 2011–2012, etc.). We resampled the input sequences with this strategy 100 times.

See S1 Text for a description of limitations of this resampling strategy.

For each of the 100 resamplings of the input sequences, we ran BEAST to sample trees, as described above, for 100 million states sampling every 10,000 states; we removed 10 million states as burn-in and resampled to obtain 1,000 sampled trees.

Then, we performed phylogeographic analyses on each of the 100 samplings of input sequences by drawing from their sampled trees. We used a discrete trait substitution model [84] on location in BEAST version 1.8.4. To estimate transition rates between locations we used a nonreversible CTMC model with $4^2 - 4 = 12$ rates. Furthermore, to evaluate the significance of routes in the diffusion process, we added indicator variables to each rate through Bayesian stochastic search variable selection (BSSVS); we set the number of nonzero rates to have a Poisson prior with a mean of 3.0, placing considerable prior probability on having the fewest rates needed to explain the diffusion. We ran BEAST with 10 million states, sampling every 1,000 states, and removed the first 1 million as burn-in. At each sampled state, we logged the complete Markov jump history [85,86], as well as a tree with the reconstructed ancestral location of each node.

To determine an MCC tree across the 100 samplings of input sequences, we ran TreeAnnotator on the sampled trees from each of the 100 samplings and then selected the MCC tree, from the 100 options, with the highest clade credibility score. We show this one, colored by reconstructed ancestral locations, in S10B Fig.

For each sampling $x_i$ of the 100 samplings of input sequences, we counted the number of jumps between each pair of locations at each state using the complete Markov jump history after resampling the jump history every 10,000 states. For each $x_i$, at each state, we calculated the fraction of migrations between each region pair by dividing the number of migrations between the pair by the total number of migrations at that state. To quantify support for migration routes in each $x_i$, we calculated Bayes factors (BFs) on the rate indicator variables. We calculated the posterior probability that a rate is nonzero as the mean of the indicator variable over the MCMC states, thereby providing a posterior odds. We calculated the prior probability that a rate is nonzero as the expected number of nonzero rates divided by the number of rates, which reduces to $1/N$, where $N$ is the number of locations; thus, the prior odds is $1/(N\text{-}1)$ or, in this case, 1/3. We set an upper limit of 10,000 on the BF and a lower limit of 1.0. To estimate the proportion of ancestry for each $x_i$, we used skyline statistics via PACT [87] to calculate proportion of ancestry at each location from each other location: in particular, for each $x_i$, we used the sampled trees with ancestral locations as input (after resampling them every 10,000 states), padded the trees with migration events, broke the trees into temporal windows of 0.1 year

going back 5 years prior to sampling, and estimated the proportion of history from tips in each time window.

To summarize phylogeographic results across the 100 resamplings $x_i$ of the input sequences, we show probability distributions across the $x_i$. When plotting the fraction of migrations to each region from each other (S10C Fig), we calculated the mean of this fraction across all the sampled states in each of the 100 runs to produce a point estimate for each $x_i$ and show the distribution of these means across the 100 $x_i$. We calculated the BFs on migration routes between the 4 regions by combining the sampled indicator variables across all 100 $x_i$ to compute the posterior odds (S10D Fig). Similarly, the proportion of ancestry plotted between a pair of locations in a time window (S10E Fig) is the mean across the 100 $x_i$ of the mean proportion for that pair in that time window from each $x_i$. We calculated the pointwise percentile bands in S10F Fig from the mean proportions in each $x_i$ across the $x_i$ (i.e., they are percentiles across the resamplings of the input sequences).

## Supporting information

**S1 Fig. Sequencing results and predictors of outcome.** (A) Distribution of mumps virus (MuV) RT-qPCR Ct value, taken at sample source, for all sequencing replicates prepared with both depletion and capture (see Materials and methods). Genome (blue): a replicate produces a genome passing the thresholds described in Materials and methods. MuV RT-qPCR serves as a predictor of sequencing outcome. (B) Distribution of collection interval (days between symptom onset and sample collection) for all samples prepared with both depletion and capture. Genome (blue) is defined as in panel A. Samples taken more than 4 days after symptom onset did not produce genomes in this study [88]. (C) Relative sequencing depth of coverage aggregated across 203 mumps genomes. (D) Number of unambiguous bases in the genome assembly of each sample by MuV:23S ratio (MuV copies by MuV RT-qPCR divided by 23S copies by 23S RT-qPCR; see Materials and methods). Each point is a replicate, colored by sequencing preparation method. (E) Normalized MuV reads (unique MuV reads divided by raw sequencing depth) in each sample by MuV:23S ratio. Points are as in panel D. Nine points with fraction mumps reads >0.04 are beyond the y-axis limits. In panels A, B, D, and E, reads from each replicate were downsampled to 1 million prior to assembly (see Materials and methods). In panels D and E, 1 point with a MuV:23S ratio $<10^{-8}$ and 3 points with a MuV:23S ratio $>10^{-3}$ are beyond the x-axis limits. Ct, cycle threshold; MuV, mumps virus; RT-qPCR, real-time quantitative polymerase chain reaction.
(TIF)

**S2 Fig. Maximum likelihood tree, root-to-tip regression, and principal component analysis.** (A) Maximum likelihood tree of the 225 mumps virus genotype G genomes used in this study. Tips are colored by sample source (MDPH or CDC); previously published genomes are indicated by unfilled circles. (B) Root-to-tip regression of genomes shown in panel A, rooted on GenBank accession KF738113 (Pune.IND, 1986). (C) Root-to-tip regression of genomes in the clade containing the two USA 2006 sequences (USA_2006; see Fig 1A) as well as their descendants. (D) Principal component analysis of genetic variants from the genomes in panel A. Each point is a genome colored by its geographic location. CDC, Centers for Disease Control and Prevention; MDPH, Massachusetts Department of Public Health.
(TIF)

**S3 Fig. Phylogenetic tree colored by institution.** MCC tree of the 225 mumps virus genotype G genomes used in this study, colored by academic institution. Clades are labeled as in Fig 1A.

MCC, maximum clade credibility.
(TIF)

**S4 Fig. Amino acid substitution in the mumps virus genome.** (A) Variation in genomes generated in this study. Each row represents one of the 119 mumps HN amino acid sequences from the individuals in our study who had known vaccine status. Samples are displayed in order of descending time since last MMR vaccine dose. Colored variants indicate variation from the consensus of all included sequences. (B) Variation in all published genotype G HN sequences. Each row represents one of the 456 publicly available mumps genotype G HN sequences (including from genomes generated in this study). Identical sequences are collapsed and then grouped by hierarchical clustering. In both panels, amino acid substitutions relative to the Jeryl Lynn vaccine strain are highlighted in blue, with orange indicating a second variant allele and green indicating a third. Light red bars indicate possible neutralizing antibody epitopes, and dark red bars indicate potential N-glycosylation sites. (C) Estimate of d$N$/d$S$ at each amino acid site in MuV coding regions, calculated across all 225 genotype G genomes used in this study. At each site, the mean estimate and 95% credible interval (not corrected for multiple testing) is shown. (D) Posterior density of d$N$/d$S$ in each gene, using the same data set. (E) MCC tree of the 225 mumps genotype G genomes used in this study, colored by vaccination status. Clades are labeled as in Fig 1A. HN, hemagglutinin gene; MCC, maximum clade credibility; MMR, Measles-Mumps-Rubella; MuV, mumps virus.
(TIF)

**S5 Fig. Comparison of PCR-positive and PCR-negative Massachusetts samples.** Comparison between 521 mumps PCR-negative samples tested in Massachusetts between 2016-01-01 and 2017-06-30, and 198 mumps samples from unique patients in Massachusetts in the same time period. In all panels, percentages have been recalculating after removing unknowns. (A) Vaccination status of positives ($n = 198$) and negatives ($n = 309$) with known vaccination status. A chi-square test suggests there is no relationship between PCR result and vaccination status ($p = 0.012$). (B) Years since vaccination for positives ($n = 198$) and negatives ($n = 309$) with known vaccination status. A chi-square tests suggests there is a relationship between PCR result and vaccination status ($p = 1.19 \times 10^{-7}$), and the plot shows that most recently vaccinated individuals were PCR-negative. (C) Collection interval for positives ($n = 196$) and negatives ($n = 477$) with known symptom onset and known collection date. A chi-square test suggests there is no relationship between PCR result and collection interval ($p = 0.918$). PCR, polymerase chain reaction.
(TIF)

**S6 Fig. Parameters used in epidemiological models.** We illustrate fitted distributions of parameters of the modeled natural history of mumps infection. (A) We calibrate a gamma distribution to the duration of the incubation period—defined from the time of mumps virus exposure to the onset of shedding—using data from experimental human mumps virus infections with known exposure times [75]. (B) Onset of mumps shedding generally precedes onset of symptoms in the clinical course. We fit a gamma distribution describing the period of latent shedding to pooled data from 2 studies [75] and (C) apply previous estimates of the distribution of the duration of shedding after parotitis onset [77]. (D) We obtain estimates of the distribution of vaccine protection within a university protection by pairing previous estimates of the association between the strength of vaccine protection and time since receipt of the last dose [5] to data on vaccine coverage in a large university [80]. (E) We infer the distribution of the generation interval length in the Harvard using data from 10 cases with known exposure sources ("contact link"). A gamma distribution fitted by maximum likelihood recovers mean

and dispersion estimates nearly identical to those reported in earlier mumps outbreaks [82].
(TIF)

**S7 Fig. Connection between epidemiological and genetic data.** (A) Multidimensional scaling applied to samples in Clade II-outbreak (see Fig 1A). Each point is a mumps genome and pairwise dissimilarities are based on Hamming distance (see Materials and methods). Genomes with known epidemiological links are connected with a red line. (B) ROC curve for samples within Clade II-outbreak using pairwise genetic distance (calculated as in panel A) as a predictor of epidemiological linkage. (C) Transmission reconstruction using individuals within Clade II-outbreak, using the SH gene or whole genome sequences. Inferred links with probability $\geq 0.7$ are shown; arrows are colored by data set. Nodes with one or more inferred links are shown and are colored by institution. See also Fig 2C, right. ROC, receiver operating characteristic; SH, small hydrophonic.
(TIF)

**S8 Fig. Trees produced with single-gene and multigene sequences.** Maximum likelihood trees using (A) the SH gene only, (B) the HN protein only, (C) a concatenation of the HN protein, the F coding region, and the SH gene, and (D) the complete mumps genome. In all panels, tips are colored by clades as defined in Fig 1A and S2 Table. The HN protein sequence does a significantly better job at capturing the epidemiologically-relevant clades than the SH gene, and the tree created from SH+HN+F (nearly 25% of the genome) closely resembles the tree created from whole genome sequences. F, fusion protein; HN, hemagglutinin-neuraminidase; SH, small hydrophobic.
(TIF)

**S9 Fig. tMRCA probability distributions for mumps genotypes using SH gene sequences.** The date of the most recent genotype A clinical sample is indicated, excluding samples that closely resemble a mumps vaccine strain. SH, small hydrophobic; tMRCA, time to the most recent common ancestor.
(TIF)

**S10 Fig. Additional analyses of global mumps spread using SH gene sequences.** (A) World map indicating number of SH sequences in our data set from each of 15 regions; the 4 circled global regions represent the 4 regions from which we resampled input for migration analyses (see Materials and methods for details regarding geographic and temporal resampling of sequences). (B) Tree with the highest clade credibility across all trees generated on resampled input from 4 global regions. Branch line thickness corresponds to posterior support for ancestry (indicated by branch color). (C) Migration between the 4 global regions shown in panel A. Each plot shows a posterior probability density, taken across resampled input, of the fraction of all reconstructed migrations that occur to the destination (indicated in upper right) from each of the other 3 sources. (D) Migration between the 4 global regions shown in panel A. Shading of each migration route indicates its statistical support (quantified with BF) in explaining the diffusion of mumps virus. (E) Average proportion of geographic ancestry of samples in each of the 4 global regions (labeled) from each of the 4 regions (colored), going back 5 years from sample collection. (F) Average proportion of Europe in geographic ancestry of US samples, and vice-versa. Shaded regions are pointwise percentile bands (2.5% to 97.5%) across 100 resamplings of the input sequences. Colors for panels B,C,D,E,F are by global region, as shown in the bottom right. BF, Bayes factor; SH, small hydrophobic.
(TIF)

**S11 Fig. Skygrid reconstruction of population size.** The scaled population size ($N_e\tau$) according to a Skygrid reconstruction. Dark line represents the median population size across samples from the posterior, and shaded area represents the 95% HPD interval. HPD, highest posterior density.
(TIF)

**S1 Table. Sample metadata.** Demographic information of all mumps cases in Massachusetts between 2016-01-01 and 2017-06-30, and the subset of these included in this study.
(TIF)

**S2 Table. Model selection and tMRCA estimates across models.** (A) Marginal likelihoods estimated in 6 models: combinations of 3 coalescent tree priors (constant size population, exponential growth population, and Skygrid) and 2 clock models (strict clock and uncorrelated relaxed clock with log-normal distribution). Estimates are with PS and SS. The BF are calculated against the model with constant size population and a strict clock. (B) Mean estimates of clock rate, date of tree root, and tMRCAs of the clades shown in Fig 1A (excluding Clade 0-HU, which consists of one sample). USA-4 corresponds to "Clades I and II" in Fig 1A. Below each mean estimate is the 95% highest posterior density interval. tMRCAs of additional unlabeled nodes are available at http://doi.org/10.5281/zenodo.3338599. BF, Bayes factor; PS, path sampling; SS, stepping-stone sampling; tMRCA, time to the most recent common ancestor.
(TIF)

**S3 Table. Viruses identified in mumps PCR-negative samples.** Influenzavirus B is the only segmented virus listed, and we identify 51 reads mapping to 6 of the 8 segments: in order, 2, 6, 0, 0, 8, 10, 4, 21 reads to each segment. PCR, polymerase chain reaction.
(TIF)

**S1 Data. Sample metadata.** Information and metadata regarding all mumps PCR-positive and PCR-negative samples on which sequencing was attempted. PCR, polymerase chain reaction.
(XLSX)

**S2 Data. Single nucleotide polymorphisms.** List of identified nonsynonymous SNPs, including the frequency of each SNP. Separate list of identified within-host variants above 2% frequency. Separate list of SNPs in possible immunogenic regions of HN and NP in US samples from 2016 to 2017 (all positions relative to the relevant gene). HN, hemagglutinin-neuraminidase; NP, nucleoprotein; SNP, single nucleotide polymorphism.
(XLSX)

**S3 Data. d$N$/d$S$ values.** Calculated d$N$/d$S$ value and confidence interval for each amino acid position across the mumps virus genome.
(XLSX)

**S4 Data. Publicly available SH and HN sequences.** List of SH and HN sequences and corresponding metadata used in analysis, as well as SH sequences available for resampling. HN, hemagglutinin-neuraminidase; SH, small hydrophobic.
(XLSX)

**S1 Text.**
(DOCX)

## Acknowledgments

We thank A. Matthews and S. Winnicki for management and guidance; I. Shlyakhter, S. Weingarten-Gabbay, S. Ye, C. Tomkins-Tinch, and other members of the Sabeti Laboratory for

discussions and reading of the manuscript; J. Hall, P. Patel, E. Buzby, K. Chen, and F. Halpern-Smith for mumps diagnosis and laboratory support; A. Osinski, C. Brandeburg, H. Johnson, J. Cohen, K. Royce, M. Popstefanija, N. Harrington, R. Hernandez, and J. Leaf for case management and epidemiological investigation; T. Mason and the Broad Institute Genomics Platform for sequencing support; M. Salit for sharing reagents. We are indebted to mumps patients and clinical and epidemiological teams for making this work possible.

The findings and conclusions in this report are those of the authors and do not necessarily represent the official position of the Centers for Disease Control and Prevention, the National Institute of General Medical Sciences, the National Institute of Allergy and Infectious Diseases, or the National Institutes of Health.

## Author Contributions

**Conceptualization:** Shirlee Wohl, Anne Piantadosi, Katherine J. Siddle, Christian B. Matranga, Sandra Smole, Yonatan H. Grad, Pardis C. Sabeti.

**Data curation:** Shirlee Wohl, Hayden C. Metsky, Meagan Burns, Rebecca J. McNall.

**Formal analysis:** Shirlee Wohl, Hayden C. Metsky, Stephen F. Schaffner, Anne Piantadosi, Joseph A. Lewnard, Bridget Chak, Lydia A. Krasilnikova, Katherine J. Siddle, Elizabeth H. Byrne.

**Funding acquisition:** Pardis C. Sabeti.

**Investigation:** Shirlee Wohl, Anne Piantadosi, Meagan Burns, Bridget Chak, Katherine J. Siddle, Bettina Bankamp, Scott Hennigan, Brandon Sabina, Rickey R. Shah, James Qu, Soheyla Gharib, Susan Fitzgerald, Paul Barreira, Stephen Fleming, Susan Lett, Lawrence C. Madoff, Sandra Smole.

**Methodology:** Hayden C. Metsky, Anne Piantadosi, Joseph A. Lewnard, Katherine J. Siddle, Christian B. Matranga, Rickey R. Shah.

**Project administration:** Shirlee Wohl, Bridget Chak, Nathan L. Yozwiak, Bronwyn L. MacInnis, Pardis C. Sabeti.

**Resources:** Paul A. Rota, Lawrence C. Madoff, Sandra Smole, Pardis C. Sabeti.

**Software:** Daniel J. Park.

**Supervision:** Christian B. Matranga, Daniel J. Park, Susan Fitzgerald, Paul Barreira, Paul A. Rota, Lawrence C. Madoff, Nathan L. Yozwiak, Bronwyn L. MacInnis, Sandra Smole, Yonatan H. Grad, Pardis C. Sabeti.

**Visualization:** Shirlee Wohl, Hayden C. Metsky, Stephen F. Schaffner, Anne Piantadosi, Joseph A. Lewnard, Elizabeth H. Byrne.

**Writing – original draft:** Shirlee Wohl, Hayden C. Metsky, Stephen F. Schaffner, Anne Piantadosi, Joseph A. Lewnard, Lydia A. Krasilnikova, Nathan L. Yozwiak, Bronwyn L. MacInnis.

**Writing – review & editing:** Shirlee Wohl, Hayden C. Metsky, Stephen F. Schaffner, Anne Piantadosi, Meagan Burns, Joseph A. Lewnard, Lydia A. Krasilnikova, Katherine J. Siddle, Christian B. Matranga, Daniel J. Park, Paul Barreira, Paul A. Rota, Lawrence C. Madoff, Nathan L. Yozwiak, Bronwyn L. MacInnis, Yonatan H. Grad, Pardis C. Sabeti.

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
