## [Editor Report · Decision Letter 0]

24 Jul 2019

Dear Dr Sabeti, 

Thank you for submitting your manuscript entitled "Combining genomics and epidemiology to track mumps virus transmission in the United States" for consideration as a Research Article by PLOS Biology.

Your manuscript has now been evaluated by the PLOS Biology editorial staff as well as by an academic editor with relevant expertise and I am writing to let you know that we would like to send your submission out for external peer review.

*Please be aware that, due to the voluntary nature of our reviewers and academic editors, manuscripts may be subject to delays during the holiday season. Thank you for your patience.*

**Important**: Please also see below for further information regarding completing the MDAR reporting checklist. The checklist can be accessed here: https://plos.io/MDARChecklist

Please re-submit your manuscript and the checklist, within two working days, i.e. by Jul 26 2019 11:59PM.

Kind regards,

Lauren A Richardson, Ph.D

Senior Editor

PLOS Biology

INFORMATION REGARDING THE REPORTING CHECKLIST:

PLOS Biology is pleased to support the "minimum reporting standards in the life sciences" initiative (https://osf.io/preprints/metaarxiv/9sm4x/). This effort brings together a number of leading journals and reproducibility experts to develop minimum expectations for reporting information about Materials (including data and code), Design, Analysis and Reporting (MDAR) in published papers. We believe broad alignment on these standards will be to the benefit of authors, reviewers, journals and the wider research community and will help drive better practise in publishing reproducible research. 

We are therefore participating in a community pilot involving a small number of life science journals to test the MDAR checklist. The checklist is intended to help authors, reviewers and editors adopt and implement the minimum reporting framework. 

IMPORTANT: We have chosen your manuscript to participate in this trial. The relevant documents can be located here:

MDAR reporting checklist (to be filled in by you): https://plos.io/MDARChecklist

**We strongly encourage you to complete the MDAR reporting checklist and return it to us with your full submission, as described above. We would also be very grateful if you could complete this author survey:

https://forms.gle/seEgCrDtM6GLKFGQA

Additional background information:

Interpreting the MDAR Framework: https://plos.io/MDARFramework

Please note that your completed checklist and survey will be shared with the minimum reporting standards working group. However, the working group will not be provided with access to the manuscript or any other confidential information including author identities, manuscript titles or abstracts. Feedback from this process will be used to consider next steps, which might include revisions to the content of the checklist. Data and materials from this initial trial will be publicly shared in September 2019. Data will only be provided in aggregate form and will not be parsed by individual article or by journal, so as to respect the confidentiality of responses. 

Please treat the checklist and elaboration as confidential as public release is planned for September 2019.

We would be grateful for any feedback you may have.

---

## [Decision Letter · Decision Letter 1]

4 Sep 2019

Dear Dr Sabeti,

Thank you very much for submitting your manuscript "Combining genomics and epidemiology to track mumps virus transmission in the United States" for consideration as a Research Article at PLOS Biology. Your manuscript has been evaluated by the PLOS Biology editors, an Academic Editor with relevant expertise, and by several independent reviewers.

As you will read, the reviewers appreciate many aspects of your study. However, they also raise some concerns that will need to be addressed in a revision. Of particular note, Reviewers #2 and #3 both believe that further evidence that vaccine escape is not contributing to the outbreak is needed. Reviewer #2 and the Academic Editor also believe that the scope of the results and title should be tempered to reflect the region where samples were obtained. 

In light of the reviews (below), we will not be able to accept the current version of the manuscript, but we would welcome resubmission of a much-revised version that takes into account the reviewers' comments. We cannot make any decision about publication until we have seen the revised manuscript and your response to the reviewers' comments. Your revised manuscript is also likely to be sent for further evaluation by the reviewers.

Your revisions should address the specific points made by each reviewer. Please submit a file detailing your responses to the editorial requests and a point-by-point response to all of the reviewers' comments that indicates the changes you have made to the manuscript. In addition to a clean copy of the manuscript, please upload a 'track-changes' version of your manuscript that specifies the edits made. This should be uploaded as a "Related" file type. You should also cite any additional relevant literature that has been published since the original submission and mention any additional citations in your response. 

Before you revise your manuscript, please review the following PLOS policy and formatting requirements checklist PDF: http://journals.plos.org/plosbiology/s/file?id=9411/plos-biology-formatting-checklist.pdf. It is helpful if you format your revision according to our requirements - should your paper subsequently be accepted, this will save time at the acceptance stage.

Please note that as a condition of publication PLOS' data policy (http://journals.plos.org/plosbiology/s/data-availability) requires that you make available all data used to draw the conclusions arrived at in your manuscript. If you have not already done so, you must include any data used in your manuscript either in appropriate repositories, within the body of the manuscript, or as supporting information (N.B. this includes any numerical values that were used to generate graphs, histograms etc.). For an example see here: http://www.plosbiology.org/article/info%3Adoi%2F10.1371%2Fjournal.pbio.1001908#s5.

For manuscripts submitted on or after 1st July 2019, we require the original, uncropped and minimally adjusted images supporting all blot and gel results reported in an article's figures or Supporting Information files. We will require these files before a manuscript can be accepted so please prepare them now, if you have not already uploaded them. Please carefully read our guidelines for how to prepare and upload this data: https://journals.plos.org/plosbiology/s/figures#loc-blot-and-gel-reporting-requirements.

Upon resubmission, the editors will assess your revision and if the editors and Academic Editor feel that the revised manuscript remains appropriate for the journal, we will send the manuscript for re-review. We aim to consult the same Academic Editor and reviewers for revised manuscripts but may consult others if needed.

We expect to receive your revised manuscript within two months. Please email us (plosbiology@plos.org) to discuss this if you have any questions or concerns, or would like to request an extension. At this stage, your manuscript remains formally under active consideration at our journal; please notify us by email if you do not wish to submit a revision and instead wish to pursue publication elsewhere, so that we may end consideration of the manuscript at PLOS Biology.

When you are ready to submit a revised version of your manuscript, please go to https://www.editorialmanager.com/pbiology/ and log in as an Author. Click the link labelled 'Submissions Needing Revision' where you will find your submission record. 

Sincerely,

Lauren A Richardson, Ph.D

Senior Editor

PLOS Biology

Reviews

Reviewer #1: Nathan D. Grubaugh, signed review

Wohl et al. provide an excellent investigation of the 2016-17 mumps virus outbreak in Massachusetts using genomic epidemiology. While I didn't have access to previous reviews and author responses, this version reads very well and tells a very interesting and complete story. The analyses all seem appropriate and the figures look nice and are mostly clear. I highly recommend this for publication and provide only a couple minor comments about the figures.

1. It wasn't until I read the main text did I really understand figure 2 (and I typically look at all of the figures before reading). There is a lot of important info buried in the legend, especially to explain B and C. Could the authors add a label to show that the left panels are estimates made from epi data and the right panels include genomic data? If this is clear from the start, the figure would be much clearer and more powerful.

2. Given the space in the main text, I would recommend brining some of the very nice and informative supplemental figures to the main text. The authors can choose, but I would recommend S1, S4 (but update the title to make it clear that this was used to investigate vaccine escape, then perhaps simplify it to showcase the main points), and S9.

--------------

Reviewer #2: 

Wohl et al. present a molecular epidemiological analysis of recent mumps outbreaks in Massachusetts during 2016-17. Using whole genome viral data, they are able to reconstruct the spread of mumps between neighboring communities including several universities in the region and connect these local outbreaks to larger-scale transmission patterns in the United States. They also explore whether mumps has undergone antigenic evolution to escape vaccine-based immunity, but do not find any genetic changes that can be correlated with vaccination patterns.

While this paper reports a perfectly valid molecular epidemiological analysis, the viral sequence dataset they analyze is limited to only ~200 samples collected mainly in MA from 2016-17. Thus authors conclusions about mumps circulation in the larger U.S. are therefore somewhat suspect, and I believe the authors over interpret what such sparsely collected data can tell them (see main criticism below) about larger circulation patterns. 

Main criticism:

1) Results Lines 102-103: The authors find that a single mumps lineage has dominated in the US since 2006 and then argue that this “implies that continuous and extensive geographic movement of mumps virus, rather than isolated outbreaks, underlies mumps in the US” —> I don’t actually see evidence for this claim in the phylogeny shown in Figure 1. If anything there is a fair amount of clustering in the phylogeny by region and lineages sampled in MA and NE one year are most closely related to lineages sampled in the those regions the previous year. It’s therefore possible that the G genotype spread slowly throughout the US and the virus is now persisting locally in each region.

2) The possibility that mutations may have allowed for vaccine escape is of major interest, but no mutations were found to strongly correlate with vaccination status or show signs of positive selection (dN/dS > 1). But classic dN/dS ratios would be very underpowered to detect selection here (Kryazhimskiy et al, PLoS Genetics, 2008) since there has presumably not been enough time for the multiple mutations needed to produce a signal of positive selection under these tests to accumulate. In light of this, maybe it would be better to plot the frequency of each mutation among sampled viruses over time and look if any mutations in antigenic regions have dramatically increased in frequency?

Minor remarks:

Line 544: ‘e-i(t)’ —> I’m guessing this should be e^{-lambda_i(t}?

--------------

Reviewer #3: Christophe Fraser, signed review

The authors generate approximately 200 new genomes of mumps virus circulating during an unusually large epidemic in and around Boston universities. They show that the virus consists of multiple widely circulating lineages. They highlight the increased resolution into transmission. Using the SH gene, they compare their sample to viruses circulating globally, and make the case for a persistent US-European lineage causing most cases. They demonstrate that the virus has an effective reproductive number RE well above one, and that control measures need to focus on blocking local transmission. A lot of work went into this impressively comprehensive analysis. 

This study is well executed. The case that there is a lot of local transmission is convincing, and the finding of multiple closely related lineages circulating is surprising.

The authors also address the question of whether the virus has mutated to escape vaccine, and conclude that waning immunity is a more likely explanation for persistent transmission. I was less convinced by the robustness of this section of the paper. I was not convinced about the power of the dN dS analysis to detect escape from the vaccine selection pressure. At least, it was not clearly motivated what assumptions or hypotheses were being tested. 

Perhaps more could be made of the data on time since vaccination, comparing negative and positive samples, and also formally testing whether subtitutions are associated with time since vaccination. 

The data sharing statement should refer both to the sequences and to the associated metadata; currently the authors only propose to share sequences. 

Specific comments:

Introductory paragraph, lines 46-56. How sure are the authors that temporally changing patterns of reporting might not affect these observations?

Line 60. ‘academic institutions and other close contact settings.’ Please use more specific language. What is a ‘close contact setting’? Does this include a household? 

Line 60, and repeated several times later. Where possible, avoid open ended ranges such as ‘at least 18’, and instead use ranges and estimates. 

Line 65. I am not convinced that ‘phylogenomic analysis’ is a thing and think ‘phylogenetic analysis’ would be clearer.

Line 76. The number of genomes is variously 201 (here), 200 (line 96), and 203 (Line 220 and Figure S1). Please check to make this consistent, or specify if slightly different samples were used for different analyses. 

Line 81. ‘Median sequencing depth’. Also specify 95% range. 

Line 81. ‘All genomes were at least 82% complete.’ This is not a fair description of the sequencing success rate, if I understand correctly the information provided in Fig S1. The authors generated 203 genomes from 259 samples. So 56 samples (22%) were excluded, because of low viral loads associated with late collection of the samples. This is relevant information for readers interested in assessing the sequencing method used here. It is also possibly relevant as a sampling bias to consider in the discussion. 

Lines 91-95. Whilst I am convinced that this conclusion is likely correct from the paper, I don’t see how you can conclude this from the analysis of the genomes (Figure 1). It rather seems to be a conclusion of the partial genes (Figure 3). To conclude this, you need to place the diversity in your sample in the context of the full global diversity, which you can only do with the SH gene. If you agree, then it seems you may need to present Figure 3 before Figure 1. I can see this is awkward since you want to showcase the genome trees & data, but you could make a case for the broad level information from a conventional SH analysis, and then zooming into the much higher resolution whole genome data. 

Lines 129-139. To me, this was the most opaque section of the paper. First on dN/dS: Why is a dN/dS analysis appropriate for addressing the question of vaccine escape? Why would vaccine selection produce higher dN than non-vaccine immunological selection? Is the study powered for this comparison? It is not clear to me that a codon-by-codon analysis, as shown in Fig S4A or B is valid without correcting for multiple testing, and I don’t understand what conclusion, if any, the authors are drawing from this analysis. In terms of the substitutions, the authors conclude that most mutations fixed here were already present in an ancestral sample (Iowa 2006), and thus that immune escape is unlikely (lines 991-998). The authors should make explicit the assumption of some kind of additive model here: why is not plausible that one of the two mutations identified, 336 and 474, could alone be responsible for vaccine escape? Without further wet-lab neutralisation assays, it must be hard to tell. Overall, I can see why the authors did the analyses, the question is interesting. But the conclusions seem vague at this stage. 

Lines 158-181. This is a convincing analysis, and really highlights the power of the whole genome data. 

Methods. The authors use well-established in-house viral genomics methods and publicly available phylogenetic software. 

BEAST analysis: could you show the estimated skygrid plots for effective population size (Ne.tau)?

PCA analysis (lines 478-481). Does this add anything to the paper?

Stochastic model. The branching process approach looks sensible. The model estimates for lines 581-586 seem interesting, but I can’t see any estimates in the paper? 

Table S1. Please include the time since vaccination for both the positive and negative samples. Could more not be made of a comparison between the two?

Lines 1005-1010. This seems an orphan paragraph. Could you add a sentence to place these findings in context?

Lines 1013-1045. These are interesting findings on lack of diversity in host, though paragraph 1035-1045 seems very speculative given the data.

---

## [Decision Letter · Decision Letter 2]

9 Dec 2019

Dear Dr Sabeti,

Thank you for submitting your revised Research Article entitled "Combining genomics and epidemiology to track mumps virus transmission in the United States" for publication in PLOS Biology. I have now obtained advice from two of the original reviewers and have discussed their comments with the Academic Editor. 

Based on the reviews, we will probably accept this manuscript for publication, assuming that you will modify the manuscript to address the remaining points raised by the reviewers. Congratulations!

We expect to receive your revised manuscript within two weeks. Your revisions should address the specific points made by each reviewer. In addition to the remaining revisions and before we will be able to formally accept your manuscript and consider it "in press", we also need to ensure that your article conforms to our guidelines. A member of our team will be in touch shortly with a set of requests. As we can't proceed until these requirements are met, your swift response will help prevent delays to publication.

*Copyediting*

*Published Peer Review History*

*Early Version*

*Submitting Your Revision*

Sincerely,

Lauren A Richardson, Ph.D 

Senior Editor

PLOS Biology

ETHICS STATEMENT:

The Ethics Statements in the submission form and Methods section of your manuscript should match verbatim. Please ensure that any changes are made to both versions.

-- Please include the full name of the IACUC/ethics committee that reviewed and approved the study protocol/permit/project license. Please also include an approval number.

-- Please include information about the form of consent (written/oral) given for research involving human participants. All research involving human participants must have been approved by the authors' Institutional Review Board (IRB) or an equivalent committee, and all clinical investigation must have been conducted according to the principles expressed in the Declaration of Helsinki.

Reviews

Reviewer #2: 

The authors' revision has improved an already good paper. But I still disagree with their major conclusion that widespread geographic dispersal must be common.

Specifically, the authors say that: "Given the modest sampling in this dataset from outside the Northeast, finding such wide geographic dispersal suggests that long-distance migration of the virus is common in the US"

I hate to be difficult, but I really don't think this is the most parsimonious conclusion to be drawn from the phylogeny in Figure 1. Rather I would say there is strong evidence for regional circulation (e.g. in MA and the NE) with occasional long-distance disperal to more distance regions of the U.S. After all, there is relatively strong clustering even from regions like the Midwest that were very under-sampled. Of course, more sampling from different regions may in fact reveal long-distance dispersal is common, I just don't feel that this should be assumed without evidence.

---------------

Reviewer #3: Christophe Fraser, signed review

The authors have done a great job responding the comments. They have strengthened their conclusions on waning immunity. They have clarified both the motivations for the different linked analyses, and highlighted limitations where appropriate. This is an impressively comprehensive study with several interesting and (to me, at least) unexpected conclusions.

---

## [Editor Report · Decision Letter 3]

3 Jan 2020

Dear Dr Sabeti,

On behalf of my colleagues and the Academic Editor, Sara Y Del Valle, I am pleased to inform you that we will be delighted to publish your Research Article in PLOS Biology. 

Early Version

PRESS 

Kind regards,

Hannah Harwood

Publication Assistant, 

PLOS Biology

on behalf of

Lauren Richardson,

Senior Editor

PLOS Biology